

# A global assessment of gross and net land change dynamics for current conditions and future scenarios

Richard Fuchs[1,2], Reinhard Prestele[1], Peter H. Verburg[1,3]

[1]Environmental Geography group, Institute for Environmental Studies (IVM), VU University Amsterdam, De Boelelaan 1085, 1081 HV, Amsterdam, The Netherlands

[2]Land Use Change group, Institute of Meteorology and Climate Research - Atmospheric Environmental Research (IMK-IFU), Karlsruhe Institute of Technology (KIT) - Campus Alpin, Kreuzeckbahnstraße 19, 82467, Garmisch-Partenkirchen, Germany

[3]Swiss Federal Institute for Forest, Snow and Landscape Research WSL, Zürcherstrasse 111, CH-8903, Birmensdorf, Switzerland

*Correspondence to*: Richard Fuchs (richard.fuchs@kit.edu)

**Abstract.** The consideration of gross land changes, meaning all area gains and losses within a pixel or administrative unit (e.g. country), plays an essential role in the estimation of total land changes. Gross land changes affect the magnitude of total land changes, which feeds back to the attribution of biogeochemical and biophysical processes related to climate change in Earth System Models. Global empirical studies on gross land changes are currently lacking. Whilst the relevance of gross changes for global change has been indicated in the literature, it is not accounted for in future land change scenarios. In this study, we extract gross and net land change dynamics from large-scale and high-resolution (30-100m) remote sensing products to create a new global gross and net change dataset. Subsequently, we developed an approach to integrate our empirically derived gross and net changes with the results of future simulation models, by accounting for the gross and net change addressed by the land use model and the gross and net change that is below the resolution of modelling. Based on our empirical data, we found that gross land change within 0.5-degree grid cells were substantially larger than net changes in all parts of the world. As 0.5-degree grid cells are a standard resolution of Earth System Models, this leads to an underestimation of the amount of change. This finding contradicts earlier studies, which assumed gross land changes to appear in shifting cultivation areas only. Applied in a future scenario, the consideration of gross land changes led to approximately 50% more land changes globally compared to a net land change representation. Gross land changes were most important in heterogeneous land systems with multiple land uses (e.g. shifting cultivation, smallholder farming, and agro-forestry systems). Moreover, the importance of gross changes decreased over time due to further polarization and intensification of land use. Our results serve as empirical database for land change dynamics that can be applied in Earth System Models and Integrated Assessment Models.

## 1 Introduction

Land change dynamics (e.g. changes in land cover or land use) play a major role in the Earth System. They have far-reaching consequences by altering many biophysical and biogeochemical ecosystem processes (e.g. albedo, greenhouse gas fluxes,



transpiration, water balance and surface roughness), which directly or indirectly drive the climate on continental to global scale (Ciais et al., 2013; Gaillard et al., 2010; Houghton et al., 2012; Shevliakova et al., 2009; Teuling et al., 2017; Zaehle and Dalmonech, 2011). Earth System Models (ESMs) are used to explore the impacts of land changes on future climate, biogeochemical cycling, and vegetation dynamics. Information on the extent and amount of land changes is usually provided

by land-use change models (LUCMs) or the land-use modules of Integrated Assessment Models (IAMs). Land-change dynamics can be provided by LUCMs and IAMS either by a 'net change approach', i.e. area gains minus area losses per grid cell, or by a 'gross change approach', i.e. area gains plus area losses per grid cell. Not accounting for gross land changes has been shown to substantially underestimate the amount of land changes and related climate effects (Arneth et al., 2017; Bayer et al., 2016; Fuchs et al., 2015, 2016; Peng et al., 2016; Prestele et al., 2016). Thus, gross changes need to be considered in

future model development.

The implementation of gross land changes faces, however, various difficulties. First, LUCMs and IAMs mostly have limited abilities to account for gross land changes at the scale of modelling. Most land-use models only account for land changes in one direction. For instance, if the model has to allocate increasing area for a specific land cover type, it is often not able to model for area losses of the same class at the same time. Second, land-use models typically simulate land changes at a spatial

resolution of 5 arcmin (ca. 10 km at the equator) or coarser, but do not account for area gains and losses happening within these grid cells. Thus, their spatial resolution is still too coarse to capture many land changes at the small scales where they occur.  Third, ESMs that implement land-change data provided by IAMs and LUCMs typically run at a resolution of 0.25-2 degrees and miss large amounts of land changes if they do not account for gross changes by aggregating from the LUCM/IAM grid to their native grid.

As part of the Coupled Model Inter-Comparison Project Phase 6 (CMIP6) many ESMs are potentially able to account for gross land changes (Arneth et al., 2017). However, empirically based gross land change data that can directly be implemented in assessment models are currently lacking on a global scale. This lack of data availability hampers a comprehensive integration of gross land change information in LUCMs and, since LUCMs often feed into ESMs and IAMs, also in ESMs and IAMs (Bayer *et al.*, 2016; Prestele *et al.*, 2017). Moreover, in recent years, the focus of assessing the impact of gross land changes

on the climate was mainly based on the historical period (Bayer et al., 2016; Fuchs et al., 2015, 2016; Hurtt et al., 2006; Wilkenskjeld et al., 2014). The role of gross land changes in future land use projections remained unclear, mostly because of the unknown magnitude of present-day gross land changes, but also the lack of understanding of how gross land change dynamics would develop with time (Arneth et al., 2017; Hurtt et al., 2011; Stocker et al., 2014). This inhibits a precise appraisal of future mitigation and adaptation potentials (Arneth et al., 2017). Currently, the Land Use Harmonization data (LUH; Hurtt

*et al.* (2011) and its updated CMIP 6 version LUH2; Hurtt et al., in prep) are the only global data sets accounting for gross land changes. However, in these datasets gross land changes are assumed to only occur in shifting cultivation areas of the tropics (Bayer et al., 2016). A global quantification of other bi-directional changes like cropland expansion and abandonment or afforestation and deforestation within grid-cell sizes of ESMs or IAMS is missing completely (Prestele *et al.*, 2017).





Empirical data, such as from remote sensing or land cover statistics, that contains information on area gains and area losses, can be used to inform LUCMs and IAMs about land changes below their native resolution (further on referred to as land changes on 'sub-pixel' scale). Such empirical data has recently become available at very high spatial resolutions (30-100 m) at continental (Bossard et al., 2000; European Environment Agency (EEA), 2006; Meiyappan et al., 2016; MoEF, 2015;

RCMRD, 2016; Roy et al., 2015; Vogelmann et al., 2001; Wickham et al., 2010, 2013) or even global scale (Jun et al., 2014). The objective of this paper is to improve the current representation of gross land changes in LUCMs and IAMs by doing an empirical analysis of gross land use changes and proposing an approach that implements empirically derived gross land changes in a global land-use model. We account for both, the gross land changes at the model scale (5-arc minute spatial resolution) and the gross land changes at sub-pixel scale. Specifically, we (1) characterize global scale relationships between

gross and net change analysing empirical data, (2) apply these findings to a future land-use change simulation, and (3) demonstrate how the consideration of gross land changes, in contrast to net land changes, can lead to substantially different results with respect to land-use composition, future land-change dynamics, and consequences for global change studies, e.g. on the global carbon cycle. Moreover, we translate the total gross and net land change into metrics that ESMs are able to use at a common resolution of 0.5 degree and thus provide a new global dataset on future gross land changes.

**2 Data and Methods**

**2.1 Empirical data**

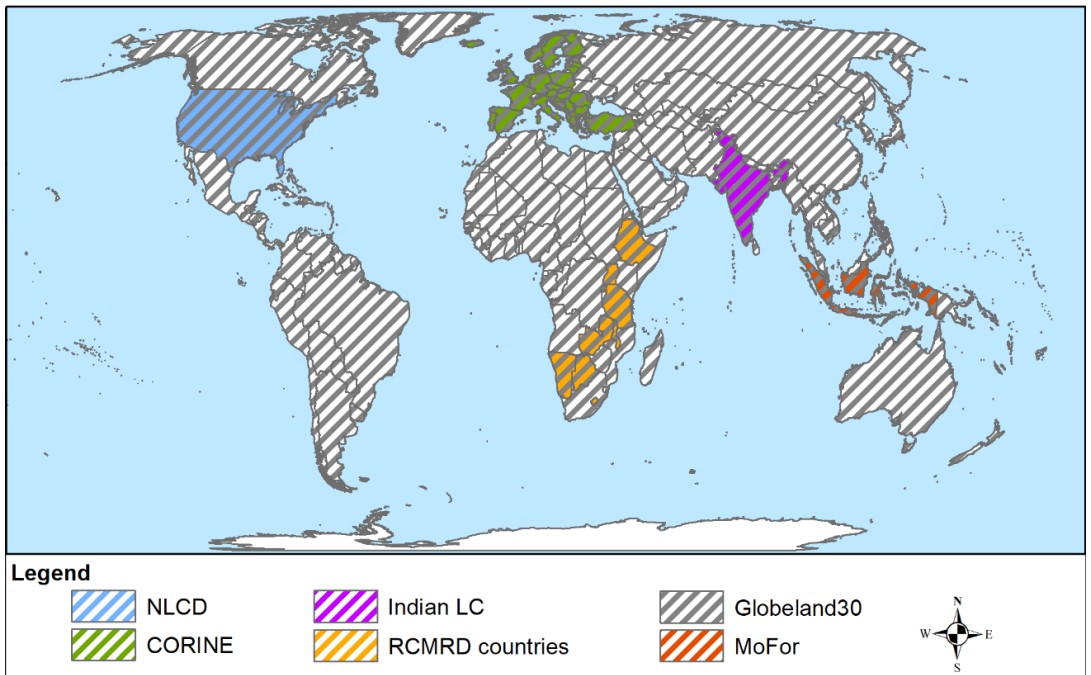

**Figure 1: Spatial coverage of high-resolution land change datasets, based on remote sensing, that were used in this study to derive gross land changes dynamics within 0.5-degree grid-cells.**



In total, we used 13 independent empirical datasets, based on remote sensing, to assess land changes on sub-pixel scale. The spatial coverage of all datasets used in this study is depicted in Fig. 1. The individual features, accuracies and available years are shown in Table 1. Since our objective was to describe future land-change dynamics, we first focused on datasets that cover

the most recent years (from 2000 onwards). Some datasets contain data for years before 2000. They were pre-processed and analysed but not used in this study. Secondly, we selected datasets that had a minimum spatial resolution of 100m, in order to account for the fine-scale land changes. However, most of the datasets that we included had a 30m resolution. Thirdly, we examined the accuracy assessments of the datasets for individual years, only including those with reasonable quality (around 80% or higher), a sufficient sampling scheme and reference data. At the time of assessment, accuracy assessments of land

changes were found to be lacking for most of the data sets. Usually, if available, the accuracy of the change products is lower than those of the individual years (e.g. see Wickham *et al.*, 2010, 2013). If an accuracy assessment was not available, we checked visually for the quality by comparing with other datasets that were available for the same larger region (Fig. 2). The accuracy of Globeland30 for the year 2010 was higher than 80% (Chen et al., 2014). However, for the year 2000, no accuracy assessment is available yet (official release was summer 2016, an extended accuracy assessment is planned). Regional accuracy

assessments of Globeland30 yielded lower accuracies, e.g. for Dar Es Salaam (Tanzania) and Kathmandu (Nepal) accuracies were 61% and 54%, respectively (Fonte et al., 2017), while in Kenya it was found to be 56-64% (See et al., 2017). Accuracies for Italy (Brovelli et al., 2015), Germany (Jokar Arsanjani et al., 2016a) and Iran (Jokar Arsanjani et al., 2016b) were greater than 78% (See et al., 2017). Although a comprehensive accuracy assessment for both years is currently not available, we decided to use this dataset for both years because of its global coverage, the high spatial resolution and reasonable quality of

land cover and changes in areas of overlap with the other datasets used in our analysis (Fig. 1). This allowed us to describe land dynamics in areas that were not covered by other datasets.

## 2.2 The CLUMondo model and the future scenario

To assess gross and net land changes for a future scenario, we used simulation output from the land system model CLUMondo (van Asselen and Verburg, 2012; Van Asselen and Verburg, 2013). This model uses a land system classification instead of the

25 more traditional land cover classification (van Asselen and Verburg, 2012). Land systems are described by a set of fractional land cover classes consisting of built-up, cropland, grassland, forest and other land, co-occurring in spatial simulation units of 9.25x9.25 km pixel. To account for regional differences, the fractional amounts of land cover per land system differ per world region. Further, land management activities such as livestock and crop production, as well as sequestered carbon are used to describe each land system. These differ for each land system per world region. A land system map for the baseline year 2000,

which is based on census and remote sensing data (van Asselen & Verburg, 2012), can be seen in Fig. 3. Land systems are allocated using local empirical relationships of land systems with explanatory biophysical and socioeconomic data, such as irrigation, population density and market accessibility (van Asselen and Verburg, 2012; Van Asselen and Verburg, 2013). Land system changes are simulated based on a demand/supply approach.



**Figure 2: Comparison of regional/continental remote sensing products (left) with global Globeland30 (right) for overlapping areas and roughly the same time spans, showing (a) CORINE land cover (2000-2012) for Greater Berlin; (c) NLCD (2001-2011) for Greater Washington; (e) Indian LC (1995-2005) for Greater Delhi; (g) RCMRD Tanzania (2000-2010) for Greater Dar es Salaam; (i) MOFOR (2000-2009) for Greater Jakarta; (b), (d), (f), (h), (j) Globeland30 (2000-2010) depicting areas of their regional/continental counterparts. Note: Cultivated land of Globeland30 comprises both cropland and pastures but is depicted in the figure with the same colour as croplands.**



Thereby, the model allocates the land systems to fulfil the demand for goods described by a scenario. The model is able to simulate gross land changes inherently by expanding a land system at one place while contracting it at another place. Within the model algorithm, each location is assigned the land system with the highest competitive power at that place. For some land

systems conversion restrictions are applied, e.g. to avoid that urban development is converted back to agricultural use and to account for conversion costs. Allowing such co-occurring area gains and losses of the same land system within a world region the model accounts for gross change dynamics at the scale of modelling. However, gross changes at sub-pixel scale are not taken into account.

In this study, we used a reference scenario for the period 2000 to 2040 to demonstrate the feasibility of our approach to include

gross land change dynamics in a land use model. This reference scenario is driven by the demand for crop production, ruminant livestock production, and the provision of built-up areas (Eitelberg *et al.*, 2016). The scenario is based on the United Nations Food and Agriculture Organization's (FAO) report: World Agriculture Towards 2030/2050, the 2012 revision (Alexandratos and Bruinsma, 2012) and characterizes the development of crop and livestock systems from 2010 to 2050. Regional-level future demands of crop production and livestock are provided by the integrated assessment model IMAGE (Stehfest et al.,

2014). Further details on the scenario can be found in Eitelberg *et al.* (2016).

## 2.3 Methods

### 2.3.1 General methods

To assess the gross or net land change dynamics at a spatial resolution relevant for ESMs and IAMs, we analysed all land changes (at the scale of modelling and sub-pixel scale) at the common spatial resolution of ESMs and IAMs (0.5-degree).

Specifically, we derived globally for every 0.5-degree grid cell an entire land change matrix containing all land-cover conversion types. We did this for each year of the simulation period. From these land-cover conversion types the specific area gains, losses as well as net and gross changes can be derived.

### 2.3.2 Land changes at the scale of modelling

In order to assess gross and net land changes at the level of land-cover types, commonly used in Earth System Models, the

land systems had to be translated back into their land cover components (e.g. grassland, cropland, forest, etc.). For the sake of efficiency, we focused on areas where land systems have changed (Fig. 4, upper left box). We then intersected all change areas with our 0.5-degree grid to calculate the land system changes per grid cell (Fig. 4, middle left box). For each of these grid cells, we converted the land systems changes into fractional land cover changes and prepared a lookup table stating the land cover area gains and losses for each class. Based on the area gains and losses per class, we derived the net and gross area

change. Subsequently, we applied the land change matrix from our empirical data analysis (see next Sect.) to derive land conversion types, e.g. from forest to grassland or from grassland to cropland (Fig. 4, upper left box). By applying the



empirically derived change matrix to the land system changes, we assume that the rates of gross change within a land system will not change in the future as long as the land system remains stable. Thus, we infer that gross changes in land cover are an inherent characteristic of the land system.

**2.3.3 Land changes at sub-pixel scale**

5 We first re-projected all original empirical datasets into an equal area projection (WGS84 Eckert IV). Subsequently, we aggregated all class legends for each product into five IPCC land categories (Intergovernmental Panel on Climate Change (IPCC), 2003): settlement, cropland (incl. orchards and agro-forestry), forest, grassland (pastures and natural grassland) and other land. In Supplement S1 – Sect. 1, we give an overview of how each legend was aggregated. The Globeland30 dataset had a class "cultivated land" that contains, besides cropland, also managed pastures that could not be separated properly. In 10 this study, we considered "cultivated land" as cropland.

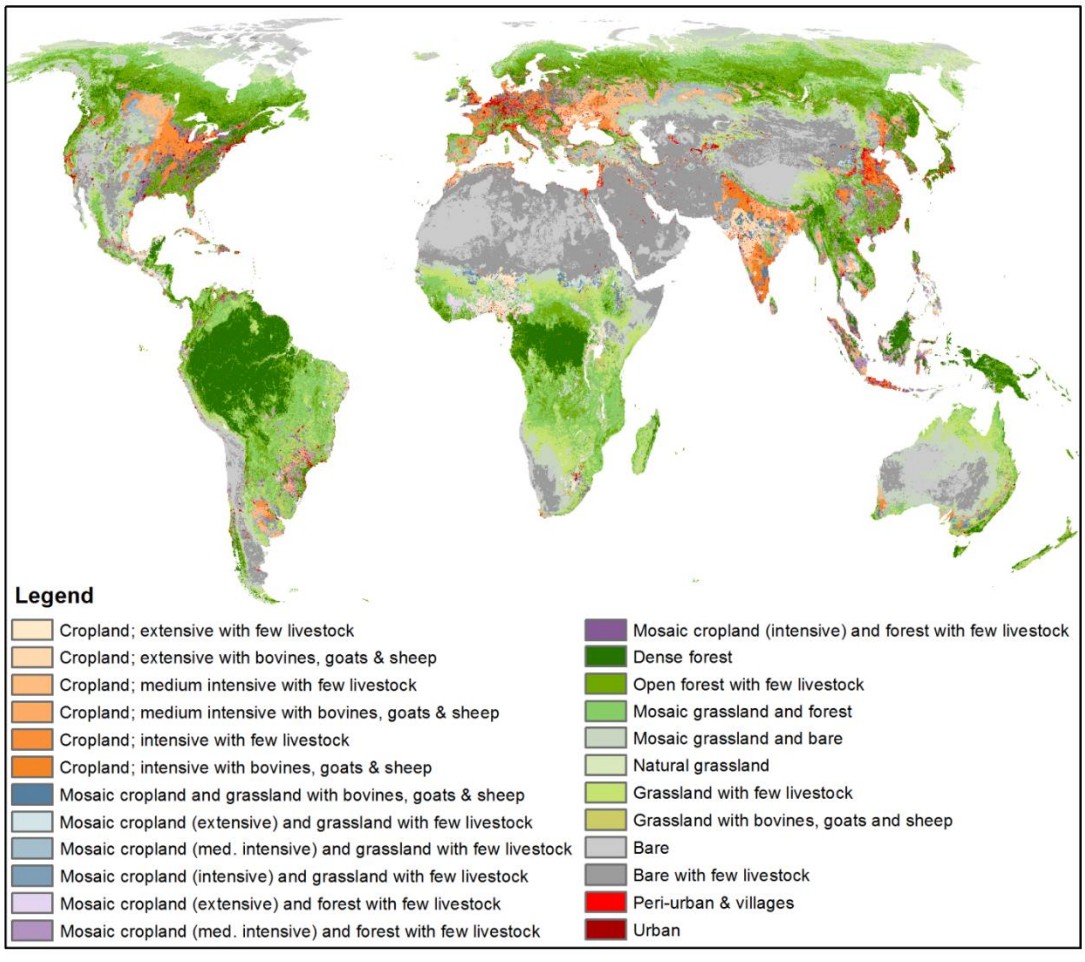

**Figure 3: Land system map for the baseline year 2000 used by the CLUMondo model.**



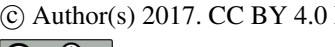

**Figure 4: Detailed overview of the approach. The approach is divided into three major steps: pre-processing (top), processing (middle) and the post-processing of the results (bottom). The left panel explains the individual steps for the analysis at the scale of modelling, using a land system model and a reference scenario for the period 2000 to 2040. The right panel shows the individual steps for the analysis at sub-pixel scale, using empirical data.**



We calculated one change dataset for every time step of each product with the original spatial resolution. For the CORINE product, we used the available change layers. For the Indian LC 1995 to 2005 and for some regions in the Globeland30, we recognized a shift by 1 pixel between the individual years when calculating land changes. This caused problems in generating the change layers. In Supplement S1 – Sect. 2, we explain in detail how we solved these problems.

We used a mask of each land system for the base year 2000 to clip our various change products by land system in order to retrieve land changes per land system. In parallel, we created a 0.5-degree grid cell layer and overlaid it with the clipped change products for each land system to create a land change matrix, containing all change areas, separately for each 0.5-degree grid cell.

The tabulation of change areas per land system within each 0.5-degree grid cell allowed us to calculate the gross/net ratio, the

net change fraction and the change matrix (Fig. 4, upper right box). The gross/net ratio explains the underestimation of changes by the net change approach compared to the gross change approach (see Fuchs *et al.*, 2015). To derive the gross/net ratio, we first retrieved the net change area (area gains minus area losses) and the gross change area (area gains plus area losses) and second we divided our gross changes by the net changes and multiplied it with 100 to get the gross/net ratio in percent. The net change fraction refers to the net change of each class as fraction of the total class area. The change matrix contains the

absolute and relative areas of all land conversion types. In order to retrieve change parameters for each land system, we averaged the gross/net ratios and change matrices of each land system for each product and time step. For the net change fraction, we took the median of all 0.5-degree grid cells per land system. The individual parameters for each land cover product can be found in Supplement S2.

From the various change products and time steps, we calculated spatially weighted averages for the change parameters to

account for the different spatial coverages of each product. In Supplement S1 – Sect. 5, we provide an overview for each dataset and its fractional contribution per land system to the final weighted average. Further, we annualized every time step of the individual products to make datasets with different time spans comparable in their change dynamics (Fig. 4, upper right box). One-year time steps are the regular time intervals of many land-use models, including the one used here.

Subsequently, we applied our derived change parameters (net change fraction, gross/net ratio and change matrix) from the

empirical study onto the reference scenario (Sect. 2.2). This allowed us to account for sub-pixel processes in the future simulation scenario (Fig. 4, middle right box). A detailed example of this procedure is found in Supplement S1 – Sect. 3.

### 2.3.4 Post-processing

In the post-processing phase, we combined our results from the land changes derived at the scale of modelling (Sect. 2.3.2) and land changes derived on sub-pixel scale (Sect. 2.3.3). We aggregated both datasets at 0.5-degree grid cell resolution (Fig.

4 bottom box) by adding the values of both data streams together. To achieve this, we rescaled land changes derived on sub-



pixel scale to 0.5-degree while keeping the sub-pixel information. One dataset was generated that contained net land changes and one that contained gross land changes. This way we could compare the differences between the two methods.

## 3 Results

### 3.1 Empirical gross land changes

In Table 2, we list all gross/net ratios that we retrieved from the empirical data for each land system, separated by land cover class. All gross/net ratios are provided as spatially weighted averages of all empirical datasets contributing to these values. Overall, the gross/net ratio of the settlement land cover ranged from 113% to 143% for the different land systems containing settlements, for cropland from 116% to 285%, for forest from 125% to 226%, for grassland from 123% to 212% and for other land from 121% to 165%. Typically, cropland, forest and grassland components have on average the highest gross/net ratios

over all land systems (149%, 149%, and 162% respectively). These classes are mostly affected by bi-directional changes, meaning gains and losses occurring at the same time within a pixel of 0.5-degrees. These changes are mostly caused by swaps of the abovementioned land cover classes, due to cultivation practices, for example shifting cultivation (temporarily cultivation of one plot, after cultivation the plot is abandoned which restores natural vegetation while a neighbouring plot is cultivated in the meanwhile). In contrast to these high gross/net ratios, settlement and other land have lower gross/net ratios (settlement

122%, other land 136%), i.e. these classes develop more one-directionally. Settlement changes are mainly characterized by urbanization. Other land is comprised primarily of inaccessible land areas (e.g. mountainous area, bare land, etc.) that rarely change and is therefore less prone to swaps with other land cover classes.

  If we separate the different land systems based on their land cover composition, it can be seen that on average homogeneous land systems have lower gross/net ratios than mosaic land systems (Fig. 5). Mosaic land systems are characterized by a large

heterogeneity of land cover classes and the spatial distribution of land cover within these land systems (indicated by the term 'mosaic' in the land system classification). Many of these land systems represent areas of smallholder farming with small parcels of land. Often these land systems have multiple land uses or perform rotational systems (e.g. shifting cultivation) that cause these higher gross/net ratios due to bi-directional changes. This is particularly the case for cropland and grassland, where the difference in gross/net ratio is the largest compared to homogeneous agricultural systems (Fig. 5). In contrast, homogeneous

classes comprise large parcels of land to minimize land management efforts and to increase land use intensity. These systems are typically more stable in terms of land changes, since production output will be achieved by intensification rather than expansion or rotation between land cover types. Therefore, the gross/net ratios of homogenous land system are lower than the ones of mosaic land systems.



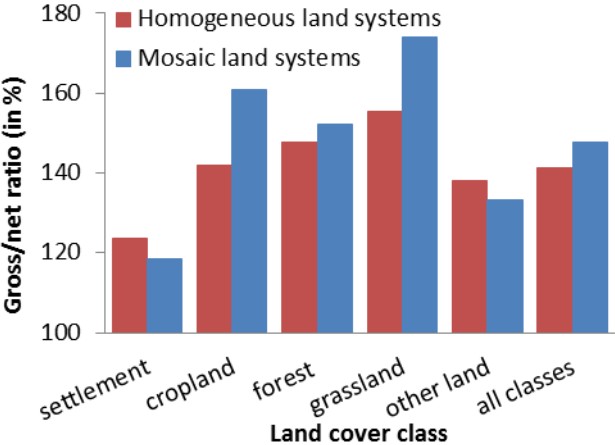

**Figure 5: Gross/net ratios for mosaic land systems (all land systems that start with 'mosaic' in their name) and homogeneous land systems (all other land systems) per land cover class and all together.**

## 3.2 Gross changes projected by the model

To demonstrate our approach for deriving gross change in future scenarios, we used a reference scenario for the period 2000-2040 based on the United Nations Food and Agriculture Organization's (FAO) report: World Agriculture Towards 2030/2050 (Alexandratos and Bruinsma, 2012) (see Sect. 2.2. for details). In Fig. 6, we show the different area gains and losses based on this scenario for each land cover component over the entire modelling period (2000-2040). The left panels (Fig. 6a, c, e, g) refer to gains and losses derived by land system changes (scale of modelling). The right panels (Fig. 6b, d, f, h) refer to the

combination of changes at both the scale of modelling and the sub-pixel scale.

Based solely on the simulated changes at model scale (Fig. 6 left panel), the main areas of land use change were found in the east coast of the United States, in Brazil and Argentina, the Sahel zone in Africa, large parts of Europe, some regions of the Middle East, India, China and Southeast Asia. Except for Eastern Europe, some parts of India, China and Mexico these changes led to widespread cropland area gains. In the East Coast of the United States, Eastern Europe, India, Argentina and Southeast

Asia, this came at the expense of forest. In Brazil, the Sahel zone, the Middle East and Northern China cropland gains occurred at the cost of grassland and other land losses. However, large parts of the world remained unaltered in the modelled scenario (see yellowish colours for each land cover class in the left panel of Fig. 6).

The combination of changes at the scale of modelling and sub-pixel scale changed the overall picture of area gain and loss (Fig. 6 right panels). Spatial patterns of changes appeared more diversified and subtle, depending on the occurrence and the

empirical parametrization of land change dynamics of each individual land system type. For instance, forest changes (Fig. 6b) appeared more widespread. Large parts of Africa and South America outside of the tropical rainforest, the boreal region, China and Australia were now subject to strong forest dynamics including reforestation

**Figure 6: Overall area gains (blue) and losses (red) per land cover class shown as change rate per year in percent for the period 2000 to 2040. Left figure panels (a, c, e, g) refer to gains and losses derived at scale of modelling. Right panels (b, d, f, h) refer to combined changes at scale of modelling and sub-pixel scale. Upper row (a, b) shows gains and loss in forest area; second row (c, d) grassland; third row (e, f) cropland; and bottom row (g, h) other land, respectively. Note: Settlement changes are not shown here due to small areas affected by changes.**



Likewise, areas of large forest losses, on the east coast of the United States and South East Asia were amplified as at the same time some reforestation is happening in these regions. Similar results were found for all other land cover classes. Additionally, the magnitude of changed area per pixel increased considerably by adding sub-pixel processes. When combining changes at scale of modelling and sub-pixel scale, the land system model and scenario implementation accounted for 20% of all gross

and net land changes, while the other 80% of changes originated sub-pixel changes. For forest and grassland, this led to larger area gains, while for cropland and other land this led to a higher magnitude of area losses. Moreover, the overall trends of gains and losses for some land-cover classes in some regions (for instance grassland in the United States) even reversed by adding sub-pixel processes (Fig. 6d) compared to the approach without including these (Fig. 6c).

## 3.3 Regional differences in accounting for net and gross land changes

In Figure 7, we added the absolute area gains and losses of all land cover classes together, comprising changes at the scale of modelling and sub-pixel scale. We depict the total net changes (Fig. 7a), total gross changes (Fig. 7b) and their difference (Fig. 7c), expressed as change rate per year and pixel in percent. Major change areas (net and gross) occurred in the eastern United States, Mexico, Colombia, Argentina, the Sahel zone, the Atlas region in northern Africa, eastern and southern Europe, Turkey, central Asia, northern India and China. The implementation of gross changes into the future scenario led to higher change

rates. While net land changes had a global average of 0.92% area change per year in this scenario, the consideration of gross land changes yielded 1.36% per year. The difference in net and gross land changes occurred mostly in large farming regions because of the scenario conditions where new agricultural areas were established (Fig. 7c). Hot spots with larger differences between net and gross land changes appeared in Mexico, Spain, Eastern Europe, parts of the Sahel zone, Central Asia, India and China mostly due to the high rates of land cover change in these regions.

In Figure 8, we illustrate the relative contribution of land cover classes to the gross change rates at modelling scale and sub-pixel scale. The individual contributions of land cover classes varied quite strongly over the whole globe. Forest changes (reddish colours) contributed most to the changes in the boreal region. Grassland changes (greenish colours) occurred most dominantly in the western United States, the Andes region, major parts of sub-Saharan Africa, Central Asia and Australia. Cropland and forest conversions (pinkish colours) can be seen in the east coast of the United States, Europe, India and South

East Asia. While at the east coast of the United States and in South East Asia the main change processes comprise cropland expansions at the cost of forests, the picture in Eastern Europe and India is the opposite (see also Fig. 6). High gross land changes of cropland expansion at the expense of grasslands (turquoise colours) occurred mostly in heterogeneous agricultural areas, like Mexico, the Sahel zone, the Mediterranean region and Northern China. These regions are known for their smallholder, mosaic land systems, which have in general a high gross change rate due to their regional land management

practice and shifting cultivation. Additionally, due to the scenario forcing, many new cropland areas were established on former grassland areas. In the wider Amazon region in Southern America, in regions around the Congo and in southern China contributions to gross land changes came from all three land cover classes (darker brownish colours).





**Figure 7: Global patterns of combined land change rates (at scale of modelling and sub-pixel scale) per year (in percent) for a reference scenario for the period 2000 to 2040. Panel (a) shows yearly change rates accounting for net land changes, panel (b) shows yearly changes rates accounting for gross land changes and panel(c) shows the difference in change rates between net and gross changes.**



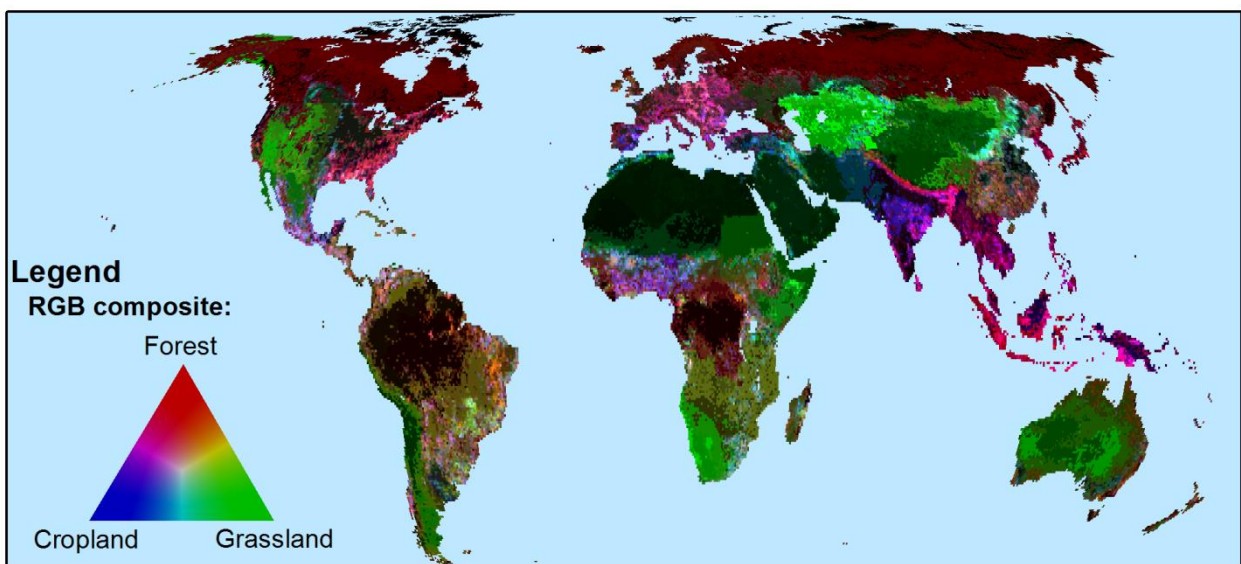

**Figure 8: Major land cover classes that cause gross land changes. Land changes comprise land changes at scale of modelling and sub-pixel changes depicted as RGB composite, with forest (red), grassland (green) and cropland (blue). Note: pink, turquoise and yellow colours refer to changes between two of these three classes (pinkish = cropland and forest; turquoise = cropland and grassland; yellowish = forest and grassland). Brighter colours refer to higher gross land changes, darker colours refer to lower gross land changes.**

## 4 Discussion

### 4.1 Evaluation of methods and processing

In this paper, we presented a first estimate of global gross land change parameters to account for gross land changes in global assessments. Our work was largely based on empirical high resolution data derived from remote sensing (30m to 100m). The individual land change products covered in total roughly 260 million km², which is an area similar to twice the entire land surface. The high spatial resolution allowed capturing land changes at scales where they occurred. This is a major advancement to previous studies. For example, within LUH (Hurtt *et al.*, 2011) the transition matrices for different land types were not based on empirical data. Within LUH2 (Hurtt et al., in prep.) only shifting cultivation was constrained by Landsat imagery using the global forest change product of Hansen *et al.* (2013). In that respect, our database is able to provide much more nuanced change dynamics for all world regions and thematic classes. We applied the land dynamic parameters in a scenario model, to demonstrate the potential of accounting for gross change in land change projections. This way, we provide the first comprehensive estimate of future gross change dynamics.



In the near future, many more datasets suitable for implementation in our approach from remote sensing can be expected. For example, the new land cover change product on a yearly basis, recently released by the Land Cover Climate Change Initiative (LC-CCI) (1992 to 2015, 300m spatial resolution) (ESA-LC-CCI, 2017). New satellites for land cover change detection were launched (e.g. Sentinel 2A and 2B, 30m spatial resolution) (ESA, 2017), which will provide a good foundation to derive new

land-cover datasets in the coming years. Even existing Landsat and MODIS satellite archives are currently used to derive new land cover change datasets which can be implemented in our approach, for example the Terra-Class datasets for Brazilian Amazonas region) (Almeida et al., 2016; Centro Regional da Amazônia, 2017).

In this study, we considered only data from the last two decades. However, additional land change data exist dating back to before the year 2000. For Africa, a few RCMRD datasets provided information for 1990 (RCMRD, 2016). The NLCD and

CORINE data also provided land-change data for the 1990's, although with lower accuracies (Vogelmann et al., 2001). The Indian LC dataset has land change data available back to 1980 (Meiyappan et al., 2016). More regional land-change datasets certainly exist. Landsat satellite archives provide data back to 1970's (USGS, 2017). For some countries, for example the Netherlands, data back to 1900 exist, which in principle allow to retrieve gross land changes (Kramer and Dorland, 2009). This historic data would allow to generate time period dependent gross/net change parameters for various world regions and

implement them in existing global historic reconstructions for recent decades to explore regional land-change dynamics (e.g. Ramankutty & Foley, 1999; Klein Goldewijk *et al.*, 2016).

Applying existing and upcoming datasets will help to further extend our database and strengthen the reliability of the land-change parameters. The use of multiple datasets for every world region would allow more robust and region specific estimates. For this study, we had to average our gross change parameters globally for each land system, due to the limited amount of data

for some regions. Averaging all empirical data sets for these land systems globally may lead to incorrect or inaccurate regional characterizations. Especially for grassland systems that occur over a very wide range of biomes (from Tundra to the Sahel zone) such averaging is not correct. Therefore, we choose to use for these different grassland systems different averages for the Northern Hemisphere and the sub-tropical grassland systems (see Supplement S1 – Sect. 4).

## 4.2 Evaluation of results and sources of uncertainty

Our empirical analysis has confirmed that gross land changes occur globally in every world region. Applied to our future reference scenario, net land changes led globally to an average of 0.92% area change per year. Based on gross land changes the average change rate was 1.35% per year, which is an increase of roughly 50% compared to the net change approach. In earlier approaches, that covered Europe only, a similar magnitude of difference between gross and net land changes could be proven outside shifting cultivation areas (Fuchs *et al.* 2015). Approximately 20% of all gross and net land changes originated

from the scenario implementation. The other 80% of changes can be explained by sub-pixel changes identified from empirical data. This points to the significance of empirical data and sub-pixel processes. Over the entire modelling period, the gross/net ratio decreased for each land cover class by 1% to 4%. This implies that gross land changes tend to play a decreasing role in later stages of the reference scenario. The main reason for this decrease was the conversion of heterogeneous, mosaic, land



systems, which were characterized by high gross/net ratios, into more homogeneous land systems, which had lower gross/net ratios (compare Fig. 5). The increasing land use intensification led to a decreasing fraction of mosaic land systems in this scenario, and therefore also to decreasing impact of gross land changes.

In the simulated gross changes and in the sub-pixel gross changes, the main areas of change were related to regions with heterogeneous land systems, such as in shifting cultivation areas of Central America, the Sahel Zone and India. Mediterranean land systems (e.g. agro-forestry) and smallholder farming systems like in China or eastern Sub-Saharan Africa also showed major changes.

The empirical data we used was subject to uncertainties as well. Although we chose datasets of justifiable data quality (e.g. high spatial resolution, large area coverage), often with an accuracy assessment, each of these datasets suffered to varying degrees from some form of misclassification. Especially in land change datasets, misclassifications from individual years add up, decreasing the overall accuracy of the change dataset. In general, areas that are affected by seasonal snow cover, droughts or temporal floods, such as wetlands, but also heterogeneous landscapes with multiple land cover components per pixel are often subject to misclassifications due to many mixed pixels that may be classified differently in separate years without an actual change. Such misclassifications are likely to lead to overestimations of the gross changes between the years. Small positional inaccuracies between years are noted as change while they are not representing change in reality. The global product, Globeland30, currently lacks a complete accuracy assessment and indeed major discrepancies with the regional/continental datasets can be seen e.g. for Africa (see Fig. 2), where accuracies from regional case studies were also reported to be lower (Fonte et al., 2017; See et al., 2017). Nonetheless, despite greater challenges for a consistent land cover classification at a global level and its inherent complexity to compete with regional/continental datasets, global datasets are able to provide us with a more comprehensive picture over the entire globe providing information also for areas that are currently underrepresented by region/continental datasets, e.g. South America and Central/East Asia.

### 4.3 Adaptation of change parameters to other legends

The approach presented in this paper used a specific future simulation model (CLUMondo) as an illustration. Other models using a different land cover class aggregation of the original classes may need to further aggregate our classes. For example, ESMs like the coupled LPJ (Smith et al., 2001) or ORCHIDEE (Ciais et al., 2005; Krinner et al., 2005) and the IAMs, like IMAGE (Stehfest et al., 2014) or MESSAGE-GLOBIOM (Havlík et al., 2014), are able to account for cropland, grasslands and forests. Urban areas and other land are considered as well, but neutral in terms of fluxes. Additionally, all these models are able to work on at least 0.5 degree and are able to account for sub-pixel processes (Bayer et al., 2016; Peng et al., 2016).

We aggregated our data to five common Intergovernmental Panel on Climate Change (IPCC) categories: settlement, cropland, forest, grassland and other land (IPCC, 2003), in order to show the potential of our approach. For each continental region, we averaged the individual land cover components across all available land systems and calculated the same land change dynamic parameter as explained in the methods. In Table 3, we show the gross/net ratios for continental regions accompanied with the datasets that went into the regrouping. The averaged land transition matrices for these regions can be found in Table 4.

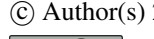



Similar to the Fig. 6, in Fig. 7 and Fig. 8 we see the highest gross/net ratios between cropland and grassland in northern and Central America, and Africa due to the shifting cultivation practice and smallholder farming systems. Highest gross/net ratios for forests could be found in North and Central America, South America and Central East Asia due to forest management practices in these areas. Compared to the independent forest cover change dataset of Hansen *et al.* (2013), we see major

hotspots of areas gains and losses appearing in these areas either next to each other or at the same spot. Other regions in Europe, Africa and South-East Asia (except Indonesia) and Australia are less affected. The gross/net ratios of Table 3, together with the transition matrices of Table 4, can be applied on existing land use scenarios (historic or future) at 0.5 degree to account for gross land change dynamics.

### 4.4 Implications for Earth System Modelling

Using our gross change data may have various implications for Earth System Modelling, since the amount of changed area determines the dynamics and quantity of carbon fluxes and the land conversion types determine on which carbon stocks the land changes have to be allocated (Bayer et al., 2016; Fuchs et al., 2016). The same applies to other biogeochemical and biophysical variables (e.g. methane, N2O, water vapour, albedo, surface roughness) (Luyssaert, 2014; Peng et al., 2016; Schulze et al., 2010; Stocker et al., 2013; Teuling et al., 2017). Global patterns of greenhouse gas fluxes will alter, depending

on the gross land change dynamics within each world region. Previous studies for Europe showed that the consideration of gross land changes altered carbon fluxes at pixel scale by up to 70% (Fuchs *et al.*, 2016). Overall, within a 60-year modelling period, the European carbon balance changed by ca. 7% when accounting for gross changes (Fuchs et al., 2016).

### Conclusions

In this study, we could show that that, based on empirical data, gross land changes occur globally in every world region. This

finding contradicts earlier studies, which assumed gross land changes to appear in shifting cultivation areas only. Applied to our future reference scenario, net land changes led globally to an average of 0.92% area change per year, while for gross land changes the average change rate was 1.35% per year. This is an increase of roughly 50% compared to the net change approach. Empirical data contributed ca. 80% of changes in the future scenario we used. This highlights the importance of accounting for sub-pixel processes in global assessments. In our scenario, gross land changes appeared in regional patterns, most dominant

in Eastern Europe, Turkey, the Sahel zone, the United States and development countries in transition, like the BRICS states (Brazil, Russia, India, China and South Africa). Large-scale and high-resolution remote sensing data was crucial for this kind of assessment. This highlights the increasing importance of land related remote sensing data in global assessments. With our approach, it is possible to further decrease uncertainties in land changes dynamics and related land atmosphere fluxes in ESMs. This again, helps to improve accuracies for future mitigation and adaptation scenarios.

### Competing interests

The authors declare that they have no conflict of interest.



## Acknowledgements

The work reported in this study was supported financially by the European Union's Seventh Framework LUC4C project and ERC Grant Agreement nr. 311819 – GLOLAND to VU University Amsterdam. We would like to thank Prof. Dr. Atul Jain, Dr. Prasanth Meiyappan, Prof. Dr. Wenbin Wu, Dr. Qiangyi Yu, Prof. Dr. Daniel Murdiyarso and Ahmad Basyiruddin Usman
for sharing their data, expertize and support with us.

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



**Table 1: Overview of high-resolution land change datasets used in this study. Note: \* refers to available years of the datasets that were not used in this study.**

| Dataset / Reference | Spatial coverage (area, name and km²) | Temporal coverage (period and years) | Reference for accuracies | Accuracies | Spatial resolution |
|---|---|---|---|---|---|
| **CORINE** | Consistent change data for max. 38 European countries (max. 5710,02 tsd km²) | 1990\*, 2000, 2006, 2012 | (Bossard *et al.*, 2000; EEA, 2007) | n/a (2000); >85% (2006); >85% (2012) | 100m x 100m |
| **NLCD** | U.S. (w/o Hawaii and Alaska) (max 7784,42 tsd km²) | 1992\*, 2001, 2006, 2011 | (Fry et al., 2011; Homer et al., 2007, 2015) | 78.70 % (2001); 78.00 % (2006); n/a (2011) | 30m x 30m |
| **Globeland30** | Global (134940,12 tsd km²) | 2000, 2010 | (Jun et al., 2014) | n/a, but planned | 30m x 30m |
| **Indian LC** | India (3287,26 tsd km²) | 1985\*, 1995, 2005 | (Meiyappan et al., 2016; Roy et al., 2015) | 94.46% (2005) n/a (1995) n/a (1985) | 100m x 100m |
| **RCMRD-Botswana** | Botswana (581,02 tsd km²) | 2000, 2010 | (RCMRD, 2016) | 92.50% (2000) 90.24% (2010) | 30m x 30m |
| **RCMRD-Ethiopia** | Ethiopia (1047,81 tsd km²) | 2003, 2008 | (RCMRD, 2016) | 87.97% (2003) 86.68% (2008) | 30m x 30m |
| **RCMRD-Lesotho** | Lesotho (30,56 tsd km²) | 2000, 2014 | (RCMRD, 2016) | 89.29% (2000) 88.73% (2010) | 30m x 30m |
| **RCMRD-Malawi** | Malawi (96,39 tsd km²) | 1990\*,2000, 2010 | (RCMRD, 2016) | 87.76% (1990) 84.88% (2000) 84.01% (2010) | 30m x 30m |
| **RCMRD-Namibia** | Namibia (825,79 tsd km²) | 2000, 2010 | (RCMRD, 2016) | 89.29% (2000) 88.73% (2010) | 30m x 30m |
| **RCMRD-Rwanda** | Rwanda (25,27 tsd km²) | 1990\*, 2000, 2010 | (RCMRD, 2016) | 82.20% (1990) 82.74% (2000) 81.30% (2010) | 30m x 30m |
| **RCMRD-Tanzania** | Tanzania (877,57 tsd km²) | 2000, 2010 | (RCMRD, 2016) | 92.50% (2000) 90.24% (2010) | 30m x 30m |
| **RCMRD-Uganda** | Uganda (209,59 tsd km²) | 2000, 2014 | (RCMRD, 2016) | 90.30% (2000) 84.22% (2014) | 30m x 30m |
| **RCMRD-Zambia** | Zambia (753,02 tsd km²) | 2000, 2010 | (RCMRD, 2016) | 89.05% (2000) 75.51% (2010) | 30m x 30m |
| **MoFor Indonesia** | Indonesia (1904,56 km²) | 1990\*, 1996\*, 2000, 2003, 2006, 2009, 2011\*, 2012\* | (MoEF, 2015; Webgis Kementerian Lingkungan Hidup Dan Kehutanan, 2017) | n/a | Initially vector format, but gridded to 30m x 30m |



**Table 2: Empirical gross/net ratios in percent for each land system, separated per land cover class components. A higher gross/net ratio indicates larger discrepancies between net and gross land change estimates (a gross/net ratio of 200 means that gross changes are double as high a net land changes). Usually, larger discrepancies occur in heterogeneous land systems, due to small–scaled bi-directional changes within the same grid-cell. Note: gross/net ratios were weighted averaged amongst all input data and temporarily normalized for one-year time steps to ensure comparability.**

| Land Systems classification | Spatially weighted average and temporarily normalized (one year) gross/ net ratio in percent | | | | |
|---|---|---|---|---|---|
| **Land system name** | **settlement** | **cropland** | **forest** | **grassland** | **other land** |
| Cropland; extensive with few livestock | 143 | 149 | 128 | 200 | 131 |
| Cropland; extensive with bovines, goats & sheep | 113 | 128 | 127 | 136 | 130 |
| Cropland; medium intensive with few livestock | 136 | 161 | 131 | 163 | 131 |
| Cropland; medium intensive with bovines, goats & sheep | 114 | 182 | 139 | 145 | 150 |
| Cropland; intensive with few livestock | 125 | 177 | 154 | 181 | 149 |
| Cropland; intensive with bovines, goats & sheep | 118 | 132 | 129 | 123 | 125 |
| Mosaic cropland and grassland with bovines, goats & sheep | 118 | 142 | 125 | 145 | 129 |
| Mosaic cropland (extensive) and grassland with few livestock | 127 | 285 | 186 | 211 | 126 |
| Mosaic cropland (medium intensive) and grassland with few livestock | 114 | 140 | 143 | 154 | 128 |
| Mosaic cropland (intensive) and grassland with few livestock | 115 | 144 | 146 | 151 | 134 |
| Mosaic cropland (extensive) and forest with few livestock | 117 | 168 | 136 | 134 | 126 |
| Mosaic cropland (medium intensive) and forest with few livestock | 116 | 131 | 137 | 156 | 128 |
| Mosaic cropland (intensive) and forest with few livestock | 114 | 147 | 150 | 194 | 165 |
| Dense forest | 119 | 132 | 146 | 160 | 152 |
| Open forest with few livestock | 123 | 150 | 168 | 161 | 138 |
| Mosaic grassland and forest | 124 | 156 | 160 | 212 | 135 |
| Mosaic grassland and bare | 122 | 136 | 186 | 209 | 128 |
| Natural grassland | 122 | 116 | 226 | 179 | 127 |
| Grassland with few livestock | 124 | 144 | 148 | 152 | 128 |
| Grassland with bovines, goats and sheep | 127 | 143 | 128 | 138 | 121 |
| Bare | 120 | 119 | 140 | 144 | 157 |
| Bare with few livestock | 126 | 133 | 185 | 149 | 149 |
| Peri-urban & villages | 126 | 136 | 135 | 139 | 139 |
| Urban | 119 | 126 | 133 | 165 | 147 |

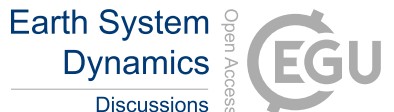

**Table 3: Regrouping of our empirical data to five IPCC land categories for continental regions as an example of adaptation potential to new legends or focus areas. The table shows region-specific gross/net ratios. High gross/net ratios indicate higher gross land changes than net land changes for different regions and classes. These values may serve as proxies for ESMs and IAMs to account for gross land changes.**

| | No. of used products | Settlement | Cropland | Forest | Grassland | Other land |
|---|---|---|---|---|---|---|
| Europe | 3 | 124 | 135 | 126 | 126 | 120 |
| North and Central America | 3 | 117 | 157 | 172 | 191 | 150 |
| South America | 1 | 141 | 133 | 160 | 154 | 153 |
| Africa | 10 | 122 | 164 | 136 | 182 | 124 |
| Central-East Asia | 1 | 117 | 133 | 188 | 159 | 154 |
| South-East Asia and Australia | 4 | 126 | 127 | 150 | 198 | 134 |





**Table 4: Regrouping of our empirical data to five IPCC land categories for continental regions as an example of adaptation potential to new legends or focus areas. The table shows averaged land transition matrices for these continental regions (conversion matrix on the left, change matrix on the right). Note: T0 refers to time step 0 and T1 refers to time step 1, indicating the direction of change in time. These values may serve as proxies for ESMs and IAMs to account for gross land changes.**

| Region | | Conversion matrix | | | | | | Change matrix | | | | | |
|---|---|---|---|---|---|---|---|---|---|---|---|---|---|
| | | settlement (T1) | cropland (T1) | forest (T1) | grassland (T1) | other land (T1) | **sum (T1)** | settlement (T1) | cropland (T1) | forest (T1) | grassland (T1) | other land (T1) | **sum (T1)** |
| **Europe** | settlement (T0) | 3,7 | 0,1 | 0,0 | 0,0 | 0,0 | **3,8** | | 2,4 | 1,0 | 1,0 | 0,4 | **4,8** |
| | cropland (T0) | 0,1 | 34,0 | 0,1 | 0,2 | 0,0 | **34,4** | 11,9 | | 7,7 | 11,2 | 1,0 | **31,8** |
| | Forest (T0) | 0,0 | 0,1 | 31,9 | 0,1 | 0,0 | **32,1** | 3,8 | 6,2 | | 3,9 | 3,1 | **17,0** |
| | grassland (T0) | 0,0 | 0,2 | 0,1 | 21,5 | 0,1 | **22,0** | 5,0 | 15,1 | 10,4 | | 5,6 | **36,1** |
| | other land (T0) | 0,0 | 0,0 | 0,0 | 0,1 | 7,5 | **7,6** | 0,5 | 0,9 | 3,5 | 5,4 | | **10,3** |
| | **sum (T0)** | **3,9** | **34,5** | **32,2** | **21,8** | **7,6** | **100** | **21,3** | **24,6** | **22,6** | **21,4** | **10,1** | **100** |
| **North and Central America** | settlement (T0) | 3,8 | 0,0 | 0,0 | 0,0 | 0,0 | **3,8** | | 0,1 | 0,1 | 0,2 | 0,0 | **0,4** |
| | cropland (T0) | 0,0 | 16,7 | 0,1 | 0,2 | 0,0 | **17,0** | 2,3 | | 1,7 | 6,1 | 1,1 | **11,2** |
| | forest (T0) | 0,0 | 0,1 | 32,7 | 1,0 | 0,1 | **33,8** | 1,5 | 1,9 | | 33,0 | 2,0 | **38,5** |
| | grassland (T0) | 0,1 | 0,2 | 0,8 | 38,8 | 0,1 | **40,0** | 3,6 | 9,5 | 21,0 | | 6,8 | **41,0** |
| | other land (T0) | 0,0 | 0,0 | 0,0 | 0,1 | 5,2 | **5,4** | 0,4 | 0,7 | 1,3 | 6,6 | | **9,0** |
| | **sum (T0)** | **3,9** | **17,0** | **33,6** | **40,1** | **5,4** | **100** | **7,8** | **12,2** | **24,1** | **45,9** | **9,9** | **100** |
| **South America** | settlement (T0) | 0,3 | 0,0 | 0,0 | 0,0 | 0,0 | **0,4** | | 0,0 | 0,1 | 0,1 | 0,0 | **0,2** |
| | cropland (T0) | 0,0 | 11,4 | 0,2 | 0,3 | 0,0 | **12,0** | 0,1 | | 2,1 | 3,7 | 0,1 | **6,1** |
| | forest (T0) | 0,0 | 0,6 | 41,0 | 2,1 | 0,1 | **43,8** | 0,1 | 7,2 | | 27,5 | 0,8 | **35,6** |
| | grassland (T0) | 0,0 | 1,0 | 2,6 | 34,1 | 0,4 | **38,1** | 0,2 | 11,3 | 31,3 | | 5,2 | **48,0** |
| | other land (T0) | 0,0 | 0,0 | 0,1 | 0,6 | 5,0 | **5,7** | 0,0 | 0,3 | 0,8 | 8,9 | | **10,0** |
| | **sum (T0)** | **0,4** | **13,1** | **43,9** | **37,1** | **5,5** | **100** | **0,5** | **18,9** | **34,2** | **40,3** | **6,1** | **100** |
| **Africa** | settlement (T0) | 0,4 | 0,0 | 0,0 | 0,0 | 0,0 | **0,4** | | 0,1 | 0,0 | 0,1 | 0,0 | **0,2** |
| | cropland (T0) | 0,0 | 6,9 | 0,3 | 1,1 | 0,1 | **8,4** | 0,2 | | 1,7 | 8,5 | 0,9 | **11,3** |
| | forest (T0) | 0,0 | 0,5 | 14,3 | 1,9 | 0,0 | **16,7** | 0,1 | 3,1 | | 12,8 | 0,2 | **16,2** |
| | grassland (T0) | 0,0 | 1,9 | 3,1 | 38,9 | 1,1 | **45,1** | 0,4 | 13,9 | 22,2 | | 16,7 | **53,2** |
| | other land (T0) | 0,0 | 0,2 | 0,0 | 1,0 | 28,2 | **29,4** | 0,2 | 2,7 | 0,4 | 15,7 | | **19,0** |
| | **sum (T0)** | **0,5** | **9,5** | **17,7** | **42,9** | **29,4** | **100** | **0,9** | **19,8** | **24,3** | **37,1** | **17,9** | **100** |





| | | | | | | | | | | | | |
|---|---|---|---|---|---|---|---|---|---|---|---|---|
| **Central-East Asia** | settlement (T0) | 1,0 | 0,0 | 0,0 | 0,0 | 0,0 | **1,0** | | 0,6 | 0,1 | 0,1 | 0,0 | **0,9** |
| | cropland (T0) | 0,1 | 21,6 | 0,1 | 0,2 | 0,1 | **22,1** | 2,5 | | 3,6 | 4,4 | 1,7 | **12,1** |
| | forest (T0) | 0,0 | 0,2 | 24,9 | 0,3 | 0,0 | **25,4** | 0,4 | 5,8 | | 8,2 | 0,8 | **15,2** |
| | grassland (T0) | 0,0 | 0,4 | 0,4 | 24,7 | 1,4 | **27,0** | 0,6 | 8,4 | 13,1 | | 16,4 | **38,5** |
| | other land (T0) | 0,0 | 0,1 | 0,0 | 2,4 | 22,0 | **24,6** | 0,2 | 3,1 | 1,0 | 29,1 | | **33,3** |
| | **sum (T0)** | **1,1** | **22,3** | **25,5** | **27,6** | **23,5** | **100** | **3,7** | **17,9** | **17,8** | **41,8** | **18,9** | **100** |
| **South-East Asia and Australia** | settlement (T0) | 0,7 | 0,0 | 0,0 | 0,0 | 0,0 | **0,7** | | 0,0 | 0,0 | 0,1 | 0,0 | **0,1** |
| | cropland (T0) | 0,0 | 12,5 | 0,1 | 0,5 | 0,0 | **13,1** | 0,7 | | 1,3 | 4,7 | 0,5 | **7,2** |
| | forest (T0) | 0,0 | 0,2 | 35,7 | 1,8 | 0,1 | **37,8** | 0,1 | 6,5 | | 32,1 | 4,2 | **42,9** |
| | grassland (T0) | 0,0 | 0,6 | 1,4 | 41,5 | 0,5 | **44,2** | 0,1 | 8,9 | 16,4 | | 9,4 | **34,8** |
| | other land (T0) | 0,0 | 0,0 | 0,2 | 0,8 | 3,2 | **4,2** | 0,1 | 0,5 | 2,9 | 11,4 | | **14,9** |
| | **sum (T0)** | **0,8** | **13,4** | **37,4** | **44,6** | **3,8** | **100** | **1,0** | **16,0** | **20,6** | **48,3** | **14,1** | **100** |