# Peer review of "A global assessment of gross and net land change dynamics for current conditions and future scenarios"

_Earth System Dynamics, 2017_

## Referee Comment (RC1) · Anonymous Referee #1 · 12 Feb 2018

It's a good paper.

-Methodology But there is no detail of method. Authors write the outline of the methodology. Please add details (equations).

-Results Almost all results are not strange, but the grass loss in the arctic seems to be high. please add the explain.

-Discussion As mentioned in section 3, uncertainty is very serious. Misclassification leads gross land use change. this is the reason why land use moder don't use middle resolution remote sensing images. Please add much more discussion.

---

## Referee Comment (RC2) · Anonymous Referee #2 · 13 Feb 2018

Review of ESD-2017-121: "A global assessment of gross and net land change dynamics for current conditions and future scenarios " The manuscript by Fuchs et al extracts the gross and net land changes using remote sensing products at the continental scale to create a new global gross and net land change dataset. Based on this dataset, authors find that the gross land changes within 0.5-degree grid cells were substantially larger than the net land changes in all parts of the world. When applied the present day gross and net land changes relationship to estimate in a future scenario, they find that the gross land changes consideration led to approximately 50% more changes globally compared to a net land change representation. The authors show that gross land changes are most important in heterogeneous land systems like shifting cultivation,

smallholder farming, and agro-forestry. This study contradicts earlier studies, which assumed gross land changes to appear in shifting cultivation areas only.

I found that the paper is well written, the results are novel and have important implications for the studies that do not consider gross land use changes. I recommend acceptance of the paper after addressing the following concerns:

1) Page 6, l25: I do not really understand what do you mean by 'intersected all changes'?

2) Fig.3 forest over India?? I am surprised to see there is no forest over Southwest coast of India (so-called Western Ghats of India)! What resolution is this data! You mention this map is based on census and remote sensing data, then I do not really understand (mostly croplands). For example see the land use land cover map for (the year 2005, 100m resolution) India (https://daac-news.ornl.gov/content/land-use-and-land-cover-india)

3) I face difficulty in understanding how you derive gross/net land changes for the future scenario at the methods section. You derive empirical relationship from observed present-day data–→then used in this empirical relationship in CLUMondo model to derive for the future scenario??? I feel figure 4 is not clear enough to convince the readers the method of deriving gross/net change ratios.

4) How do you deal with very small fractions in the denominator while calculating gross/net ratio? Worth mentioning in the discussion section.

5) Worth mentioning 'how you estimate the accuracy of the datasets'.

6) Could you provide expansion of the 'LNCD', CORINE, RCMRD, MOFOR in the caption of Figure 1?

7) At 2nd line in the first paragraph of page 9: expand LC

typo: remove repeated 'that' in the first line of Conclusions section.

---

## Author Comment (AC1) · 19 Mar 2018

Comment_R1#1 It's a good paper.

Response_R1#1 We thank the reviewer for the positive and constructive comments and suggestions. Please see the detailed point-by-point responses below.

Comment_R1#2 -Methodology But there is no detail of method. Authors write the outline of the methodology. Please add details (equations).

Response_R1#2 We will expand the methods by more detailed explanation and where needed mention the equations in explicit or descriptive form. However, in order to

stay within a reasonable page limit we will add some of the methods detail to the supplementary material.

Comment_R1#3 -Results Almost all results are not strange, but the grass loss in the arctic seems to be high. please add the explain.

Response_R1#3 This is a valid point. We discussed this already in the initial manuscript version and the supplementary material (Supplement S1 – Sect 4.). The grass loss in the artic involve land cover changes from grasslands to other land. We are aware that these particular changes are potentially too high. Some of the grass-land system classes occur in Tundra and semi-arid regions (e.g. Sahel). While the semi-arid regions are known for high land cover conversions rates, the Tundra region is not. Since we averaged all empirical changes amongst land system classes the high conversion rates of the semi-arid regions "spilled over" to the Tundra regions as well. We tackled this problem already in the methodology and will add a paragraph and some additional explanation in the supplement of the original document. However, the reviewer is correct; some overestimations for the Tundra remain. We will now clearly indicate in the discussions that an overestimation for this particular region may remain and that this should be considered in applications using our data.

Suggested change: "Nonetheless, an overestimation of this particular region, the Tun-dra, may remain. This should be taken into account in applications using our data." (pg. 17, line 23).

Comment_R1#4 -Discussion As mentioned in section 3, uncertainty is very serious. Misclassification leads gross land use change. this is the reason why land use moder don't use middle resolution remote sensing images. Please add much more discussion.

Response_R1#4 The reviewer is right that misclassifications in the empirical datasets may lead to an overestimation of gross land use change in some areas. We will elab-orate more on this in the discussion. Although, in this paper, we present a collection of the most recent and advanced datasets available for land cover applications, uncertainty is an essential part of these datasets. Contrary to many other available products, we comprehensively document uncertainties in the underlying data. In that sense, we are able to account for uncertainty, while for many other products, without an accuracy assessment, this is not possible. In the methods and discussion section, we will expand the description of the different sources of uncertainty to inform the readers as best as possible and to enable them to judge the quality of the individual products.

Suggested change: "Especially in land change datasets, misclassifications from individual years add up, decreasing the overall accuracy of the change dataset. This may affect the magnitude of gross changes in our scenario by increasing the gross/net ratio." (pg.18, line 13).

––––––––––––––––––––––––––––––

---

## Author Comment (AC2) · 19 Mar 2018

Comment_R2#1 Review of ESD-2017-121: "A global assessment of gross and net land change dynamics for current conditions and future scenarios " The manuscript by Fuchs et al extracts the gross and net land changes using remote sensing products at the continental scale to create a new global gross and net land change dataset. Based on this dataset, authors find that the gross land changes within 0.5-degree grid cells were substantially larger than the net land changes in all parts of the world. When applied the present day gross and net land changes relationship to estimate in a future scenario, they find that the gross land changes consideration led to approximately 50%

more changes globally compared to a net land change representation. The authors show that gross land changes are most important in heterogeneous land systems like shifting cultivation, smallholder farming, and agro-forestry. This study contradicts earlier studies, which assumed gross land changes to appear in shifting cultivation areas only.

I found that the paper is well written, the results are novel and have important implications for the studies that do not consider gross land use changes. I recommend acceptance of the paper after addressing the following concerns:

Response_R2#1 We thank the reviewer for the positive and constructive comments and suggestions. Please see the detailed point-by-point responses below.

Comment_R2#2 1) Page 6, l25: I do not really understand what do you mean by 'intersected all changes'?

Response_R2#2 We will remove this sentence to avoid confusion.

Comment_R2#3 2) Fig.3 forest over India?? I am surprised to see there is no forest over Southwest coast of India (so-called Western Ghats of India)! What resolution is this data! You mention this map is based on census and remote sensing data, then I do not really understand (mostly croplands). For example see the land use land cover map for (the year 2005, 100m resolution) India (https://daac-news.ornl.gov/content/land-use-and-land-cover-india)

Response_R2#3 We agree with the reviewer that it is hard to see in the image. The region mentioned by the reviewer is mostly classified as forest mosaics in the land system map (purple colors), which consist of a considerable amount of forest in this region (>30%). Given that many mountain ridges are often grasslands rather than forest, the LS map represents this as extensive mixed classes. The forest fractions in the map are based on the 500m Vegetation Continuous Fields product by Hansen et al. (2003). Grassland and cropland fractions were derived from Ramankutty et al. (2008)

at 5-arc min. spatial resolution. The approach for the LS map is explained in detail in van Asselen and Verburg (2012), Global Change Biology.

Comment_R2#4 3) I face difficulty in understanding how you derive gross/net land changes for the future scenario at the methods section. You derive empirical relationship from observed present-day data–! Then used in this empirical relationship in CLUMondo model to derive for the future scenario??? I feel figure 4 is not clear enough to convince the readers the method of deriving gross/net change ratios.

Response_R2#4 We will indicate more clearly in figure 4 that the model gross/net changes are used at the spatial resolution of the model (9.25x9.25 model) and the empirical data were used to derive changes at the sub-pixel spatial resolution through more consistency in terms. For the scenario, we assumed that for a particular land system historically observed changes would be valid also for future dynamics within this land system.

Suggested change: We changed figure 4 accordingly.

Comment_R2#5 4) How do you deal with very small fractions in the denominator while calculating gross/net ratio? Worth mentioning in the discussion section.

Response_R2#5 Good point! We will add some explanation on that in the text.

Suggested change: "Occasionally it happened that the net change fraction was very small and let to very high gross/net ratios. When these small fractions of net change led to a gross/net ratio larger than 1000% we excluded these numbers from our analysis." (pg. 18, line 8).

Comment_R2#6 5) Worth mentioning 'how you estimate the accuracy of the datasets'.

Response_R2#6 We will make clearer that we did not estimate the accuracy of the datasets ourselves. The available accuracy assessments were made by the individual institutions in their data documentation. If no assessment was available at all (e.g. for Globeland30), we focused on available case studies (page 4). Additionally, we

highlighted a visual comparison of all overlaying data set (figure 2) to compare general patterns of land cover classes.

Suggested change: "Thirdly, we used available accuracy assessments of the datasets for individual years, made by the individual institutions, only including those with reasonable quality (around 80% or higher), a sufficient sampling scheme and reference data." (pg.4, line10)

Comment_R2#7 6) Could you provide expansion of the 'LNCD', CORINE, RCMRD, MOFOR in the caption of Figure 1?

Response_R2#7 Will be included.

Comment_R2#8 7) At 2nd line in the first paragraph of page 9: expand LC typo: remove repeated 'that' in the first line of Conclusions section.

Response_R2#8 Will be changed accordingly.